# SILAC-Based Characterization of Plasma-Derived Extracellular Vesicles in Patients Undergoing Partial Hepatectomy

**DOI:** 10.3390/ijms251910685

**Published:** 2024-10-04

**Authors:** Ulrike Resch, Hubert Hackl, David Pereyra, Jonas Santol, Laura Brunnthaler, Joel Probst, Anna Sofie Jankoschek, Monika Aiad, Hendrik Nolte, Marcus Krueger, Patrick Starlinger, Alice Assinger

**Affiliations:** 1Department of Vascular Biology and Thrombosis Research, Centre of Physiology and Pharmacology, Medical University of Vienna, Schwarzspanierstrasse 17, A-1090 Vienna, Austria; 2Cluster of Excellence/Cellular Stress Responses in Aging/Associated Diseases (CECAD), Proteomics Core Facilities, University of Cologne, Joseph-Stelzmann Strasse 26, D-50931 Cologne, Germany; 3Institute of Bioinformatics, Biocenter, Medical University of Innsbruck, Innrain 80, A-6020 Innsbruck, Austria; 4Department of Surgery, General Hospital, Medical University of Vienna, Währinger Gürtel 18-20, A-1090 Vienna, Austria; 5Department of Surgery, Division of Hepatobiliary and Pancreatic Surgery, Mayo Clinic, Rochester, MN 55902, USA

**Keywords:** liver regeneration, extracellular vesicles, proteomics, post-hepatectomy liver failure

## Abstract

Post-hepatectomy liver failure (PHLF) remains a significant risk for patients undergoing partial hepatectomy (PHx). Reliable prognostic markers and treatments to enhance liver regeneration are lacking. Plasma nanoparticles, including lipoproteins, exosomes, and extracellular vesicles (EVs), can reflect systemic and tissue-wide proteostasis and stress, potentially aiding liver regeneration. However, their role in PHLF is still unknown. Methods: Our study included nine patients with hepatocellular carcinoma (HCC) undergoing PHx: three patients with PHLF, three patients undergoing the associating liver partition and portal vein ligation for staged hepatectomy (ALPPS) procedure, and three matched controls without complications after PHx. Patient plasma was collected before PHx as well as 1 and 5 days after. EVs were isolated by ultracentrifugation, and extracted proteins were subjected to quantitative mass spectrometry using a super-SILAC mix prepared from primary and cancer cell lines. Results: We identified 2625 and quantified 2570 proteins in the EVs of PHx patients. Among these, 53 proteins were significantly upregulated and 32 were downregulated in patients with PHLF compared to those without PHLF. Furthermore, 110 proteins were upregulated and 78 were downregulated in PHLF patients compared to those undergoing ALPPS. The EV proteomic signature in PHLF indicates significant disruptions in protein translation, proteostasis, and intracellular vesicle biogenesis, as well as alterations in proteins involved in extracellular matrix (ECM) remodelling and the metabolic and cell cycle pathways, already present before PHx. Conclusions: Longitudinal proteomic analysis of the EVs circulating in the plasma of human patients undergoing PHx uncovers proteomic signatures associated with PHLF, which reflect dying hepatocytes and endothelial cells and were already present before PHx.

## 1. Introduction

The liver is the largest solid organ in the body, fulfilling essential functions involving the synthesis, metabolism, and distribution of proteins, lipids, metabolites, and hormones via secretion into the blood. The liver’s unprecedented capacity to regenerate, already described in Ancient Greek mythology, acts as a rheostat to compensate for tissue injury caused by an unfavorable lifestyle or environmental exposure to toxic compounds, for example. However, chronic tissue damage gives rise to liver cancer, which is the sixth most common cancer worldwide. For these patients, partial hepatectomy (PHx) is the primary therapeutic approach, but insufficient hepatic regeneration can lead to post-hepatectomy liver failure (PHLF), which remains a significant clinical concern associated with patient morbidity and mortality [1]. Most liver hypertrophy strategies manipulate portal blood flow, including portal vein ligation (PVL), portal vein embolization (PVE), and ALPPS (associating liver partition with portal vein ligation). However, the exact mechanisms underlying hypertrophy and the balance of risks and benefits for these techniques remain unclear [2]. Despite recent advances, prediction tools and prognostic markers that could be used to reliably predict clinical outcomes following surgery are lacking [3,4]. During the past decade, research on the molecular mechanisms of liver regeneration gradually uncovered a highly complex and tightly regulated sequence of events necessary for tissue repair [5], and a range of regenerative growth factors and mediators have been identified. These are released from the liver tissue itself or from myeloid cells in response to surgical intervention. An unknown proportion of secreted molecules are packed into micro-nanosized (40 nm up to 1 μm) lipid membrane-encircled particles, termed extracellular vesicles (EVs). An increasing number of bioanalytical studies show that EVs are charged with proteins, nucleic acids, and metabolites, reflecting the donor cell phenotype at the particular condition of their release [6,7]. Being released into the extracellular space and further into blood circulation, EVs fulfill important immunological and tissue homeostatic functions by mediating communication between cells (immune, parenchymal, stromal) and tissue, distant from their origin [8]. Their cargo is increasingly recognized to infer disease-specific processes, including cardiovascular, metabolic, neuropathogenic, and oncogenic signatures [9,10,11,12,13] as well as ageing [14]. Building on these studies, the therapeutic application of EVs is actively being explored [15], including as a treatment option for liver failure [16]. To date, 57 studies have demonstrated that EVs and their cargo serve as reliable biomarkers and effective treatment options, at least in animal models and in in vitro systems.

Despite the growing interest in EV research, an unbiased longitudinal proteomic analysis of EVs circulating in human patients’ plasma following PHx has not been described so far. Anticipating inter- and intra-patient heterogeneity, we utilized a custom-made super-SILAC mix to compare EVs from patients stratified on the basis of (i) post-surgical outcome and (ii) sampling time, with the aim to identify etiologic factors promoting liver regeneration. In addition, we compared EVs from ALPPS and non-ALPPS patients to identify specific EV cargo that potentially drives liver regeneration in ALPPS patients.

## 2. Results

### 2.1. Study Overview and Patient Characteristics

We included nine patients with hepatocellular carcinoma (HCC) subjected to PHx who were analyzed in regard to PHx outcome (PHLF or noPHLF, ALPPS) and in regard to sampling time (prior to surgery, as well as on postoperative days 1 and 5). All patients (seven male, two female, average age 62.9 years) had elevated perioperative liver enzymes (GGT, AST, ALT) and a variety of comorbidities (steatosis 60%, steatohepatitis 33.3%, fibrosis 66.6%), and all received neoadjuvant chemotherapy prior to surgery. A detailed breakdown of the patient group’s characteristics can be found in Appendix A and a scheme of the study design and sample processing workflow is given in Figure 1.

### 2.2. Qualitative Proteome Signatures of Plasma EVs

The initial estimation of proteome quality and complexity was performed using SDS-PAGE profiles and indicated that albumin (70 kDa) was prominently present in the EVs. Individual protein identifications of each sample were made with SDS-PAGE gels, and the common and unique proteins identified in the respective groups (PHLF, noPHLF, ALPPS) are shown in Venn diagrams (Figure 2A). Overall, we identified 2625 (LFQ-L) and 2570 (SILAC) proteins across all the samples, with an average of 50% missing values in each sample, as illustrated in a heatmap (Appendix A). A detailed analysis of LFQ-L versus SILAC-only proteins in regard to plasma proteins is described in the Section 4.5 (Appendix A). GOCC categorization of all identified EV proteins, illustrated in a pie chart, indicated a higher proportion of nuclear proteins (29%) and a small fraction of lysosomal proteins (1%), while other compartments were comparably (10–13%) present in EVs, as shown in Figure 2B, and their percentual share of the total intensity is depicted in Appendix A. We evaluated the contributions of blood plasma proteins, categorized into complement (25), acute phase proteins (12), coagulation and fibrinolysis (13), and lipoproteins (15), as well as typical Vesiclepedia-denoted proteins (81), in regard to the overall EV protein intensities. This analysis showed that (i) the intensity of blood proteins contributed, on average, 1% of the total intensity, except for complement (2%); in addition, the intensity for Vesiclepedia proteins was highlighted (8%) and (ii) their abundance levels fluctuated the most before and 5 days after PHx (Appendix A).

Since proteins detectable in only one condition are not amenable to comparative statistical analysis, we analyzed the overlaps of these unique “only” proteins in each group (considering the outcome alone or in combination with time) beforehand. The proteins present in the EVs, specifically in PHLF, noPHLF, or ALPPS, irrespective of sampling time, are condensed in Venn diagrams (Figure 2C) with their localization and functions categorized on the basis of GOCC and Uniprot-Keywords, as summarized in Figure 2D and specified in Appendix A. Nuclear proteins were prominent in both PHLF and noPHLF+ALPPS EVs (85/71), followed by extracellular vesicles (72/43); mitochondrial (20/25), cell membrane (25/20), and Golgi apparatuses (7/9); endoplasmic reticulum (ER); and endosome (15/18), peroxisomal (4/2), and secreted (10/15) proteins. Noticeably, lysosomal proteins (CTSB, GAA, CTSA, VAMP7, and LAMTOR1) were only present in PHLF EVs. Functional classification revealed that hydrolases, proteases, kinases, and proteins with transport functions were prominent in PHLF EVs (37/19, 15/7, 15/10, 19/23, and 36/20). Proteins with *(paracrine)* signal functions present in PHLF EVs (19) included leukocyte (ALCAM/CD166), T-cell, endothelial, and platelet activation cell surface glycoprotein CD109, the regulator of extracellular glutathione, GGT1 [17], or MCFD2, which is essential in the transport of the coagulation factors FV and FVIII [18] and described to be secreted in EVs in HCC [19], for example. The proteins assigned with signal function in noPHLF and ALPPS (22) included CD14, CD47, ITGB3/CD61, and ITGA2/CD49B cell surface markers; the hepatocyte growth factor-like protein MST1; the TGFB-induced protein TGFBI; selenoprotein P (SEPP1); and the Vitamin K-dependent coagulation regulator PROZ, presumably reflecting the originating cell type. We found a noticeable number of proteins implicated in cellular stress response, encompassing cell death pathways (37 in PHLF and 32 in noPHLF +ALPPS) and autophagy and mitophagy pathways (9 in PHLF, 10 in noPHLF+ALPPS), and few proteins associated with senescence (4 in PHLF, 3 in noPHLF +ALPPS), together representing cellular programs in various cell types present in the liver that are critically relevant to maintaining homeostasis, summarized in Venn diagrams in Appendix A. These included apoptotic (14/16), necroptotic (20/15), and ferroptotic (6/3) proteins which may act as the respective pathways at different levels and fine-tune death versus survival decision programs. In PHLF for example, we identified the proapoptotic DDX47 and PAWR (PRKC apoptosis WT1 regulator protein), as well as the antiapoptotic Bax inhibitor 1 (TMBIM6). In noPHLF and ALPPS, apoptosis executioner proteins ACIN1 and TIAL1 (Nucleolysin TIAR), the ferroptosis mediator MTCH1 (Mitochondrial carrier homolog 1), and the antiapoptotic PDCD4 and CISD2 (CDGSH iron sulfur domain 2) proteins were present in EVs. Notably, ER-lipid raft proteins ERLIN1 and ERLIN-2 were only present in noPHLF EVs. An integrative analysis of all “only” proteins revealed unique and overlapping molecular signatures (MSig) in PHLF, noPHLF, and ALPPs EVs, including Fatty Acid Metabolism, Myc Targets V1/V2, DNA repair, mTORC1 Signaling, or Oxidative phosphorylation, for example, as depicted in Figure 2E. Details on the proteins in the respective pathways are summarized in Appendix A, and the enriched protein–protein interaction network modules are shown in Appendix A.

The overlaps of the “only” proteins shown in the Venn diagrams in Figure 2A were further compared between outcome groups separately for the three timepoints, with analysis results provided in Appendix A. Before PHx, 86, 67, and 136 EV proteins were unique to PHLF, noPHLF, and ALPPS, respectively. On postoperative day 1, 54,150, and 45 EV proteins were unique to PHLF, noPHLF, and ALPPS, and 5 days after PHx, 60, 99, and 93 were unique to PHLF, noPHLF, and ALPPS. The proteins with the highest LFQ-L abundance (Top5) are listed in a table. For example, PRE-PHLF EV-specific proteins included coagulation factor F12, the proinflammatory cytokine IL18, and STAT1, a key regulator in interferon signaling. Mannan-binding lectin serine protease 2 (MASP2) and Ficolin-2 (FCN2), coordinately known to activate the lectin complement pathway, were only found on postoperative day 1 in PHLF EVs, and high mobility group box proteins HMGB1 and 2 were only found in PHLF EVs on postoperative day 5. The proteins only detected in noPHLF EVs prior to surgery included KEAP1, ADAMTS13, BMP1, and the adenylyl cyclase inhibitory Gai subfamily of heteromeric G-proteins (GNA1,2,3). In noPHLF EVs on the first day after PHx, oxidative stress-reducing protein SOD2, gluthathionperoxidase1 (GPX1), and the prenylcysteine oxidase-like PCYOX1, recently shown to be adipocyte-specific EV cargo, [20] were unique, while the tyrosine protein kinases YES, SRC, and LYN and the antioxidant selenoprotein Thioredoxin reductase 1 (TXNRD1) were unique to PHLF EVs 5 days after PHx. The unique proteins detected in PRE-ALPPS EVs included the hepatocyte growth factor-like proteins MST1, MASP2 and the small GTPases RAP1A/1B, as well as the secretion-associated GTPases SAR1A/1B. Proteins implicated in oxidative stress response, including FTH1, Deoxyribose-phosphate aldolase (DERA), and Argininosuccinate synthase (ASS1), were detected in ALPPS EVs 1 day after surgery. The hypoxia-associated protein Stonin-2, the calcium-binding and alarmin protein S100A7, and the procoagulant Kallikrein KLKB1 were detected in ALPPS EVs 5 days after PHx only and were shown to be present in EVs earlier [21,22,23,24].

### 2.3. Quantitative Proteome Signatures of Plasma EVs

Principle component analyses (PCA) on the EV proteomes in regard to outcome illustrated a higher heterogeneity in noPHLF and ALPPS as compared to PHLF samples (Figure 3A). Following the analysis scheme outlined in Figure 3B, pairwise comparisons of PHLF versus noPHLF and ALPPS were performed. Differentially expressed proteins (DEPs) were visualized in volcano plots, with proteins meeting the significance criteria (conservative *p*-value < 0.05/−log_10_ > 1.3, no multiple testing correction) color coded as shown in Figure 3C,D. In these analyses, 53 proteins were upregulated and 32 proteins were downregulated in PHLF versus noPHLF, while in comparison with ALPPS, 110 proteins were higher in PHLF and 78 proteins were lower in ALPPS-EVs. 

Of all significant proteins, we found 14 proteins (APOE, HNRNPA0, HSP90AB1, HSPB1, LAMP1, MAP4, NAA50, PDHB, RUVBL1, SDHA, TUBB, TUBB2A; YWHAG, YWHAQ) commonly up- and 6 proteins (ATP2A2, CLTC1, EEF1B2, NACA4, RPL27A, TIMM44) commonly downregulated in PHLF, as illustrated in a heatmap (Figure 3E). Accordingly, PHLF EVs display an increased abundance of metabolic-TCA-cycle-core proteins, including pyruvate dehydrogenase beta subunit PDHB, providing acetyl-CoA from glycolysis to TCA, and the mitochondrial respiratory chain component succinate-ubiquinone oxidoreductase SDHA, while the abundance of the mitochondrial inner-membrane-to-matrix translocase TIMM44 was reduced, suggesting that EVs reflect mitochondrial dysfunction. We also found an increased abundance of phagosome components (TUBB, TUBB2A, and LAMP1) as well as proteins involved in the mitotic cell cycle (MAP4, NAA50, RUVBL1). The proteins downregulated in PHLF EVs included the elongation factor EEF1B2, which is involved in the transfer of aminoacylated tRNAs to the ribosome during translation elongation; RPL27A, a protein synthesis component of the 60S ribosome; and NACA, a protein that binds to nascent polypeptides lacking a signal peptide to prevent mistranslocation to ER. There was also a reduced abundance of the sarcoplasmic ATPase ATP2A2, which is involved in transporting cytosolic calcium to ER, altogether indicating disturbed translation, proteostasis, and likely an unfolded protein response (UPR), which may be further aggravated through abnormal vesicle biogenesis due to the reduced abundance of the coated pits and vesicle component CTLTC1, for example. A pathway enrichment analysis considering all the PHLF EVs’ 149 upregulated and 104 downregulated proteins after merging the comparisons against noPHLF and ALPPS is shown in the bubble plots in Figure 3F,G. For example, the proteins involved in oxidative and chemical stress response (APOE, CAPN2, GSR, LONP1, PRDX2, TXN, ERO1A), hemostasis (C1QBP, MIF, TF, ATP1B1, CAPZA2, AK3, and tubulins) mitochondrial protein degradation (HSPD1, IDH3A, OXCT1, PDHB, SSBP1), or cell cycle (MCM5, PCNA, RAD21, RPA1, MAPRE1) were higher in PHLF EVs. The proteins assigned to neutrophil degranulation (ANXA2, CTSB, VCP, ATP11A), liver development (ACO2, ASS1, CAD, GMPS) or VEGFA VEGFR2 signaling (DAPZB, LRRFIP2, UBAP2L, KATNAL2) were lower in the PHLF EVs. A complete list of the enriched GO terms with the proteins involved can be found in Appendix A.

Of note and in line with our qualitative analysis of EV cargo (Venn diagrams in Figure 2C and Appendix A), we found distinct quantitative differences when comparing PHLF EVs with noPHLF and ALPPS EVs, as evident from the volcano plots in Figure 3C,D, but also when comparing noPHLF and ALPPS EVs, as further elaborated in the GO and PPI network enrichment analyses shown in Appendix A. In these analyses, PHLF EVs were enriched in signatures assigned to the pathways “TP53 regulates metabolic genes” and “Translocation of SLC2A4 (GLUT4) to the plasma membrane” when compared to noPHLF and ALPPS EVs, respectively. In comparisons of noPHLF versus ALPPS, “Platelet aggregation” and “Protein kinase A signaling” were higher in ALPPS. Altogether, omitting sampling time as a parameter in pairwise comparisons provided additional insight into the biological processes potentially affected in PHLF patients, including ER stress, compromised protein translation, impaired (intracellular) vesicular transport, and mitochondrial dysfunction.

### 2.4. Presurgical EV Signatures of Patients with PHLF

In earlier qualitative EV analyses we found 86 proteins only present in PHLF EVs prior to surgery (Appendix A). Consequently, we finally focused on the quantitative differences found prior to surgery and compared PHLF versuswith noPHLF and ALPPS, and the PCA plots showed a clear separation, despite the limited sample size (*n* = 3), Figure 4A. Following a comparative statistical analysis, 17 proteins were higher and 12 proteins were lower in PHLF versus noPHLF, while 29 proteins were higher and 11 lower in PHLF versus ALPPS (Appendix A. Only 2 proteins (Heat shock protein beta-1 (HSPB1) and 14-3-3 protein beta/alpha (YWHAB)) were commonly higher in PHLF and 1 protein (60s ribosomal protein L27a (RPL27A)) was lower in PHLF versus noPHLF and ALPPS. In subsequent pathway enrichment analyses, we included “only” proteins (Appendix A). Accordingly, PHLF EVs carry signatures assigned to “Metabolism of RNA”, “Axon guidance”, or “Regulation of proteolysis”, while noPHLF and ALPPS EVs are enriched in proteins assigned to “Hemostasis”, “Translation”, “Protein maturation”, “Wound healing”, or “Protein localization to organelle”, for example (Figure 4B,C). Taken together, these data hint towards a PHLF-specific EV signature that is already present prior to surgery.

## 3. Discussion

In a SILAC-based proteomic analysis of nine HCC patients undergoing PHx, we found that patients who developed PHLF exhibited distinct EV proteome signatures, even before surgery. Our analysis revealed significant dysregulation in pathways related to protein translation, proteostasis, intracellular vesicle biogenesis, metabolism, and cell cycles in PHLF-derived EVs. Specifically, proteins involved in mRNA surveillance pathways, the spliceosome, and cell cycle regulation were notably altered, suggesting underlying issues in the translation machinery and vesicular transport processes crucial for cellular homeostasis and tissue regeneration.

The study highlights significant differences in the protein compositions of EVs between PHLF patients and those without PHLF who were treated with ALPPS, indicating distinct cellular and functional characteristics. PHLF EVs are enriched with lysosomal proteins and those involved in hydrolase, protease, and kinase activities and transport functions, indicating altered cellular processing and signaling in PHLF.

The presence of specific signal proteins in PHLF EVs points to potential paracrine signaling roles contributing to liver pathology and failure. In this regard, we confirmed our recent finding of a role of HMGB1 in PHLF [25], which we found in PHLF EVs only. Moreover, and as already reported in previous studies characterizing EV proteins in HCC, we confirmed that GGT1 [17] and MCFD2 [18] were present only in PHLF EVs in this study, and we speculate that in addition to the macroscopic grading of HCC, patient-specific HCC-molecular signatures may be present in EVs. Differences in apoptosis-related proteins between groups suggest that the regulatory mechanisms of cell death vary between PHLF and noPHLF/ALPPS conditions, as indicated from recent in vitro works [26].

Additionally, ER-residing lipid raft proteins ERLIN1 and ERLIN2, which respond to sterol depletion and promote autophagosome formation, were found exclusively in noPHLF EVs. This suggests that noPHLF EVs may play a role in counteracting lipid imbalances and enhancing autophagy, contributing to better cellular homeostasis and recovery post-hepatectomy.

Interestingly, EV proteome heterogeneity was higher in noPHLF and ALPPS samples compared to PHLF. Pathway analysis of PHLF proteome signatures highlighted enrichment in cell cycle regulation, vesicle biogenesis, RNA metabolism, and ER stress response, suggesting that specific EV proteomic signatures can predict PHLF outcomes, characterized by ER stress and impaired vesicular transport. The proteomic profile of PHLF-derived EVs suggests that they might serve as facsimiles of dying hepatocytes and endothelial cells due to the presence of proteins associated with cell death and stress responses in PHLF-derived EVs, contrasting with EVs from regenerative processes that support cell survival, proliferation, and differentiation. Interestingly, these pathological signatures are apparent before PHx. Whether or not EVs could serve as predictive tools to determine patients at risk of developing PHLF needs to be determined in further studies. The complex interplay of a variety of cell types (re)acting in regenerating liver tissue, including hepatocytes and cholangiocytes as well as mesenchymal, endothelial, and immune cells, has recently been resolved at the transcriptional level with single-cell resolution [27]. Each of these cell types naturally secrete proteins packed with EVs in response to stress and/or adaption by unconventional secretion or following cell death. When we overlayed our data with these single-cell transcriptomics data, we found 17 genes (15 hepatocytes and 2 immune cells) with significantly higher expression in embolized liver tissue presenting proteins exclusive to PHLF EVs. At the same time, we also found 88 genes assigned to regeneration across all cell types in the “225-only” PHLF EVs. Furthermore, 19 genes (14 hepatocytes, 5 immune cells) assigned to embolized liver and 68 genes (mostly endothelial cells and cholangiocytes) assigned to regenerating liver were overlapping with 198 proteins only detected in noPHLF and ALPPS EVs. To what extent the transcriptional landscape corroborates the circulating plasma EV proteome will be the subject of future follow-up studies.

Our study is limited in sample size and, thus, statistical power. We decided to omit the imputation of missing values and instead undertook qualitative analysis after careful filtering. By longitudinally combining samples obtained from individuals, we were able to diminish inter-subject variabilities. From a methodological point of view, our study was limited to ultracentrifugation-based EV purification due to the limited amount of plasma samples, in which co-purification of abundant blood proteins, including lipoproteins or complements, for example, is frequently observed [28]. Our study is unique in regard to the SILAC-spike-in methodology employed, which enables comparative quantitative proteome analysis of an unlimited number of experimental conditions or clinical samples, eliminating artificially introduced systematic errors before (protein digestion, cleanup), during (chromatography, mass analyzer), and after measurement (data analysis). On the other hand, a DDA-dependent acquisition for peptide/precursor quantification is necessary, which relies on sufficiently high intensity signals of peptides and, thus, limits the depth and data completeness. As anticipated, typical blood/plasma proteins are not amendable to SILAC labelling. This limitation was alleviated by analyzing the proteins’ natural (LFQ-L) signals in this study as well. Nonetheless, the number of identified plasma EV proteins identified in this study exceeds all DDA-based proteomic studies so far considering the limited sample volume used, reaching identification numbers currently achievable only with DIA methodologies. In recognition of the continuous improvements in MS-instruments and algorithms, SILAC-based quantification also becomes attractive in DIA-based proteomics. With the advent of personalized medicine, we face a growing interest in biomarker signatures that are able to describe the complexities of disease mechanisms and etiologies. Hence, high-throughput proteome analysis of both tissue and liquid biopsies—such as EVs, for example—will definitely take place in the future and may also lead to a revival of the SILAC methodology, which is, compared to currently used diagnostics (i.e., ELISA, flow cytometry), both cost-efficient and comprehensive.

Taken together, SILAC-based proteomic characterization of plasma EVs in patients undergoing PHx provides novel insights into the proteomic changes associated with PHLF. The signature of EVs in patients with PHLF indicates significant disruptions in protein translation and vesicle biogenesis, and mirrors that of dying hepatocytes and endothelial cells, which contrasts with the traditional role of EVs in promoting regeneration in other contexts.

## 4. Materials and Methods

### 4.1. Patient Cohort

Patients with HCC were recruited at the Clinic Landstrasse, Vienna, between January 2016 and March 2019 [29]. Blood was taken pre-surgery, as well as one and five days after surgery. Demographics, laboratory parameters, and PHLF (using ISGLS) were assessed [30]. Informed consent was obtained, adhering to ethical guidelines, and registered on ClinicalTrials.gov (NCT01921985, NCT01700231; accessed on 5 July 2024). Morbidity was defined using the criteria put forth by Dindo et al. [31]. Severe morbidity was defined as grade 3 or higher. PHLF was graded based on criteria established by the International Study Group of Liver Surgery. The diagnosis of PHLF was characterized by elevated serum bilirubin (SB) levels and prolonged prothrombin time (PT) persisting on the fifth day after surgery. For patients with abnormal SB and PT levels before surgery, SB had to increase and PT decrease compared to preoperative values to indicate PHLF. Patients who did not undergo routine postoperative blood draws due to good clinical condition or early discharge were classified as not having PHLF.

This study included 3 patients with PHLF, 3 patients undergoing the associating liver partition and portal vein ligation for staged hepatectomy (ALPPS) procedure [26], and 3 matched controls without complications (noPHLF). CTAD-anticoagulated plasma was collected from the portal vein 1 day perioperatively as well as 1 and 5 days after resection of malignant tissue, sequentially centrifuged at 1000× *g* and 10,000× *g* at 4 °C for 10 min to ensure removal of all cellular components [32], stored in 15 μL aliquots at −80 °C until further use, and carefully deposited in a biobank at −80 °C.

### 4.2. Isolation of Plasma EVs by Ultracentrifugation

For EV isolation, 110 μL plasma was diluted with 1mL PBS and centrifuged at 100,000× *g*/41,000 rpm100.000× *g*/41.000 rpm using a fixed angle TLA45 rotor and an Optima TLX Ultracentrifuge (Beckman Coulter) for 45 min at 16 °C. Supernatants were removed by pipetting, EV pellets were washed by resuspending in 1 mL PBS and centrifuged again. EV pellets were resuspended in a final volume of 50 μL PBS, 10 μL was used for SDS-PAGE-based protein content, quality, and complexity estimation, and 30 μL was subjected to methanol-assisted protein precipitation and delipidation. For this, 10 volumes (500 μL) of cold methanol was added and samples were vortexed and sonicated for 1min, incubated overnight at −20 °C, and centrifuged for 20 min at 13.000× *g* and 4 °C. The protein pellet was washed through the addition of 500 μL methanol. After centrifugation, pellets were air-dried and processed for proteomic analysis. The reproducibility of ultracentrifugation-based EV isolation for LC-MS/MS analysis had been proven beforehand with plasma from 3 subjects (á 150 μluL in replica) without SILAC-spike, and the LC-MS/MS analysis results are shown in Appendix A).

### 4.3. Super-SILAC and Sample Preparation for Mass Spectrometry

A heavy isotope-labelled super-SILAC-proteome standard was prepared from a variety of cell types and cultured in SILAC-heavy DMEM, as described previously [33]. In detail, the HCC cell lines HepG2, Hep3B, and Huh7; colorectal cancer cell line HCT116, prostate cancer cell line LnCAP, and leukemia; monocytic cell lines HL-60 and U937; the EAhy 926 hybridoma (HUVEC/A549) cell line displaying stable endothelial characteristics; and primary HUVEC and HAEC were cultured in lysine- and arginine-depleted DMEM supplemented with 10% dialyzed fetal bovine serum and 73 mg/L heavy [^13^C_6_^15^N_2_) L-lysine and 28 mg/L [^13^C_6_^15^N_4_] L-arginine (all Silantes), as well as 100 U/ml penicillin/streptomycin (Invitrogen). Cells were grown for at least five generations to allow full incorporation of labelled amino acids. Individual cell lysates were prepared in a lysis buffer (4% SDS in PBS) and SILAC labelling was checked after FASP digestion [34] and MS analysis, as described below. The super-SILAC mix was prepared from a mixture of all cells’ lysates and 100μg methanol-precipitated aliquots, which were stored at −80 °C until use. EV protein pellets were solubilized in 6 M urea and 2 M thiourea in HEPES pH 8.0 and mixed with 10μg super-SILAC mix, dissolved in the same buffer. After reduction (10 mM DTT) and cystein alkylation (55 mM CAA), proteins were sequentially digested with LysC (0.5 μL 0.05 mg/mL, 3 h RT), diluted with 50 mM Ammoniumbicarbonate to 2 M Urea, and incubated with Trypsin (1 μL 0.05 mg/mL) overnight. Finally, digests were acidified to 1% Formic Acid (FA), desalted, and concentrated using C18 stage-Tips [35].

### 4.4. LC-MS/MS Analysis and Raw Data Processing

Peptides were analyzed on a nano UHPLC 1000 (Proxeon) coupled via a nanoelectrospray ionization source to a Q-Exactive^TM^ Plus Hybrid-Quadrupol Orbitrap^TM^ Mass Spectrometer (Thermo Fisher Scientific, San José, CA, USA) operating in a top 12 data-dependent mode (DDA), as described, but using a 2 h gradient [36]. Raw files were initially processed with MaxQuant (v 1.5.3.8, 2016) and (v2.1.3.0 2024) with MS/MS spectra assigned to the human Uniprot database (UP000005640_9606, 20.590 entries) [37]. Default settings for mass tolerance and peptide length were used and enzyme specificity was set to LysC/P and Trypsin/P. Identification and SILAC quantitation settings were set to a min. ratio count of 1, and matching between dependent peptides, as well as iBAQ and LFQ quantitation, was enabled to quantify peptides with a missing MS1/2, to account for human plasma proteins absent in our super-SILAC mix, and to evaluate the original abundance of plasma EV proteins. N-term acetylation, oxidation (M), and phosphorylation (STY) were set as variable, and cystein carbamido-methylation and N-terminal protein acetylation as fixed modification.

### 4.5. Data Analysis

The MaxQuant output (proteingroups.txt) was analyzed using Perseus (v.1.6.15.0) software [38]. SILAC (normalized H/L ratios) and LFQ-L were uploaded and filtered for reverse, potential contaminants, keratins, and immunoglobulins (66/90); LFQ-L and SILAC ratios (L/H) were log2 transformed; data distribution was inspected in histograms; and data completeness was visualized in heatmaps (Appendix A). Sample groups were annotated (outcome, time, and the combination thereof) and proteins were annotated with “lipoproteins”, “complement”, “coagulation”, “acute phase”, “fibrinolysis”, “serpins”, and Vesiclepedia Top100, as well as with Gene Ontology terms for cellular component (GOCC) and Mitocarta 3.0. Anticipating that plasma-specific proteins might be absent in the super-SILAC proteome standard (no SILAC ratio) and appear as missing values (NaN), we compared LFQ-L (2626) and SILAC (2570) protein identifications in Venn diagrams and found that 117 LFQ-L “only” proteins constituted proteins involved in complement activation, immune effector functions, and ECM regulation, displaying liver hepatocytes, Kupffer cells, and lung dendritic signatures, while 62 SILAC-only proteins were enriched in functions involving cell cycle regulation, late endosome to vacuole transport, and antiviral response and displayed neuronal cortical stem cell, skeletal muscle, and bone marrow pro B cell signatures, as illustrated in the Venn diagrams and enrichment plots in Appendix A. GOCC categorization of the proteins in the “outcome” and “time” groups is summarized in Appendix A. To evaluate the robustness of our SILAC proteome standard, the variance of 3506 proteins with LFQ-H identified across all the samples was found to be 2.4 ± 1.5%, while only 153 proteins had variations higher than 5% (range 5–40.6%) and 11 proteins had a variation higher than 10% (CP, EP400, IGM, IGHM, C5, VWF, NRAS, BPIFC, TMSB10, FGA, FGB, Appendix A). These 153 proteins were enriched in biological functions such as Eukaryotic Translation Elongation (23 proteins, log(p): −32), Platelet degranulation (13 proteins, log(p): −12.9), ER-Phagosome pathway (10 proteins, log(p): −10.5), mRNA processing (11 proteins, log(p): −7.2), and VEGFA VEGFR2 signaling (15 proteins, log(p): −9.8). Qualitative comparisons of the protein identifications in the outcome groups were filtered for at least 3 valid values (LFQ-L) in the outcome groups to scrutinize proteins consistently identified only in PHLF or noPHLF/ALPPS, then categorized based on localization and function using Uniprot-Keyword, GOCC, KEGG, Reactome, and published datasets [39,40,41,42,43,44]. For statistical analyses considering outcome and time, proteins with SILAC ratios were filtered for at least 2 valid values in each group. SILAC ratios were centered through median z-score normalization, and pairwise statistical comparisons were performed using two-sided Student’s T-tests using conservative (*p* < 0.05) as well as permutation-based estimations of false discovery rates (250 permutations, FDR cut-off q < 0.05). Protein set enrichment analyses were done with the open-source tools Metascape [45], Enrichr-KG [46], and GSEA [47,48,49]. Graphical visualizations were done in Instant Clue [50], SRplot [51], Cytoscape v3.10.1 [52], GraphPad Prism (v.8.0), and Excel.

## Figures and Tables

**Figure 1 ijms-25-10685-f001:**
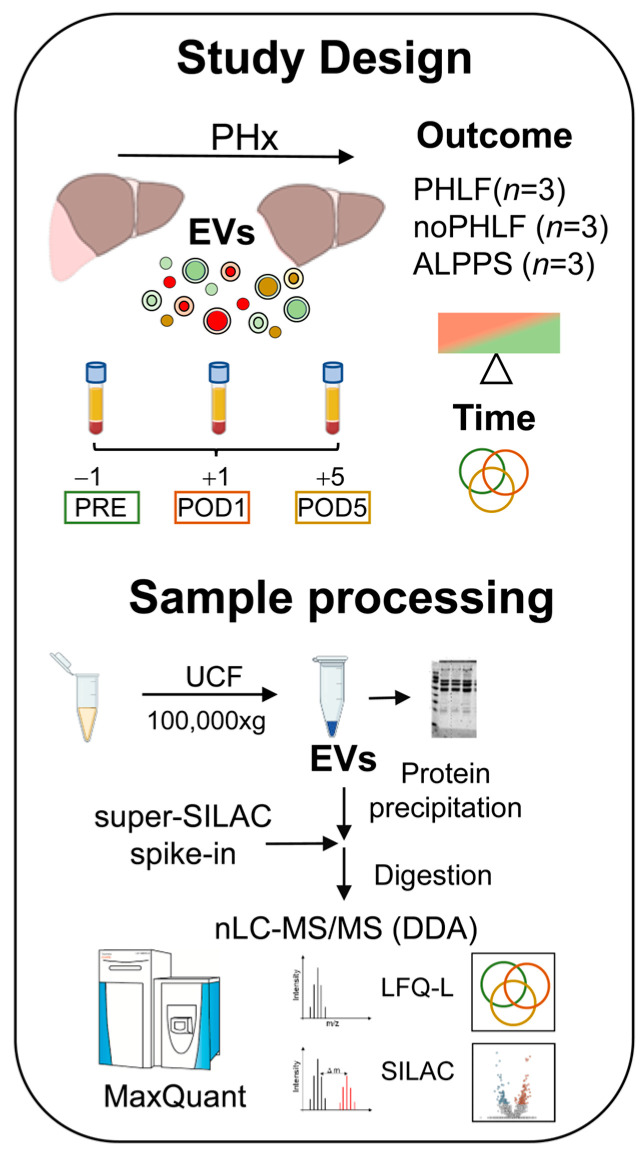
Study design and workflow of sample processing for proteomic analysis of plasma EVs.

**Figure 2 ijms-25-10685-f002:**
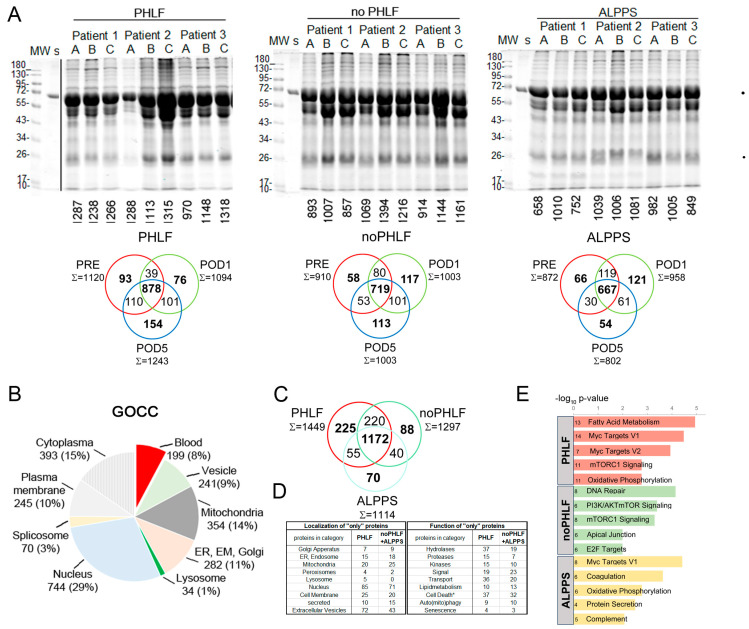
Qualitative proteome signatures of plasma EVs: (**A**) SDS-PAGE protein profiles, with numbers of identified proteins in LC-MS analysis denoted to individual samples (A = before PHx, PRE; B = 1 day after PHx; C = 5 days after PHx) and Venn diagrams showing common and unique proteins identified in respective sample groups (PHLF, noPHLF, ALPPS) after filtering for at least 2 valid values on basis of LFQ-L intensities. (**B**) GOCC categorization of all identified proteins. (**C**) Venn diagram of overlapping and “only” proteins identified in outcome groups PHLF, noPHLF, and ALPPS, irrespective of sampling time, after filtering for at least 3 valid values (LFQ-L intensities). Corresponding source data can be found in Appendix A. (**D**) Localization and functional categorization of proteins identified only in PHLF (225), or noPHLF and ALPPS (198), on basis of Uniprot-Keywords, KEGG, Reactome, and literature. * Category of cell death, encompassing apoptosis, necroptosis, and ferroptosis. (**E**) Top five enriched molecular signatures of proteins only identified in PHLF, noPHLF, or ALPPS, with −log10 *p*-Values on x-axis and numbers of proteins in each category denoted. Corresponding source data can be found in Appendix A.

**Figure 3 ijms-25-10685-f003:**
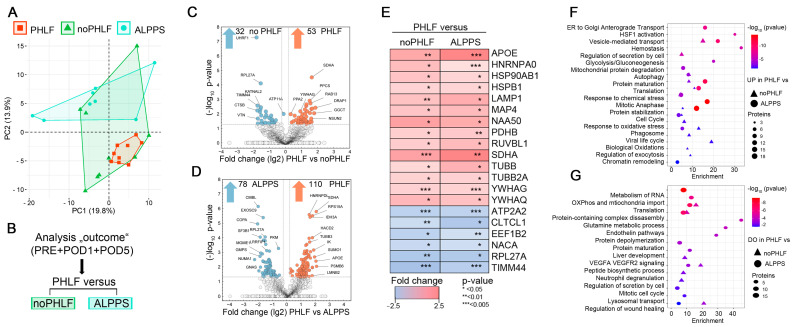
Quantitative proteome signatures of plasma EVs: (**A**) Unsupervised PCA scaled to unit variance of proteins identified in all samples (296) with outcome groups indicated. (**B**) Analysis of workflow for statistical analysis of differentially expressed proteins (DEPs) on basis of SILAC ratios. (**C**,**D**) Volcano plots depicting DEPs considering conservative *p*-value cutoff (*p* < 0.05/−log10 *p* > 1.3) as significant. (**E**) Heatmap of commonly up- and downregulated proteins in pairwise comparisons of PHLF versus noPHLF and ALPPS. (**F**,**G**) Bubble plot depicting enriched pathway terms denoting PHLF EV proteins significantly (**F**) upregulated or (**G**) downregulated in comparison to both noPHLF and ALPPS. Source data can be found in Appendix A.

**Figure 4 ijms-25-10685-f004:**
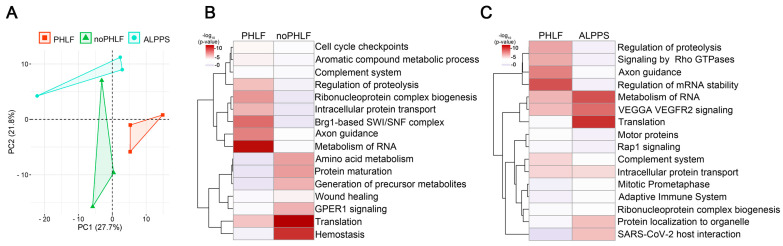
EV signatures in PHLF patients prior to surgery: (**A**) PCA of EV protein cargo before PHx in outcome groups PHLF, noPHLF, and ALPPS. (**B**,**C**) Heatmap summarizing enriched GO terms of up- or downregulated and “only” proteins in (**B**) PHLF versus noPHLF and (**C**) PHLF versus ALPPS. Source data can be found in Appendix A.

## Data Availability

All data needed to evaluate the conclusions in the paper are present in the paper and/or the Appendix A.

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
