# Peer review of "SILAC-Based Characterization of Plasma-Derived Extracellular Vesicles in Patients Undergoing Partial Hepatectomy"

_ijms, 2024, doi:10.3390/ijms251910685_

Round 1

Reviewer 1 Report

Comments and Suggestions for Authors

Study evaluating the predictive capacity of vesicular proteomics in patients undergoing partial hepatectomy. Although the technique of purification, concentration of Evs, and sequencing is innovative and the qualitative and quantitative analysis is very well detailed, some minor comments could improve the quality of the manuscript.

1.     In supplementary table 1, the etiology of the liver disease of the 9 patients included and the BCLC classification of each of them should be clear and specified. It is not clear and it is something important to describe.

2.     Among the regulatory mechanisms of cell death, only the vesicular proteins linked to apoptosis are discussed. I would like to know if you have identified any vesicular protein linked to ferroptosis, senescence, necrosis, autophagy?

3.     The first paragraph of the discussion can be completely eliminated, it is repetitive of what was discussed about EVs in the introduction. The discussion should begin with the main findings of the manuscript, paragraph two.

4.     A great limitation is the very small N, however the great strength of the study is the application of the SILAC methodology. It would be important to add to the strengths the time required by this type of methodology compared to traditional ones and finally hypothesize in the discussion about the translation into clinical practice, possibilities or how far we are.

5.     Overall, the spelling and grammar are incorrect, so we recommend that you get the English proofread.

Comments on the Quality of English Language

Study evaluating the predictive capacity of vesicular proteomics in patients undergoing partial hepatectomy. Although the technique of purification, concentration of Evs, and sequencing is innovative and the qualitative and quantitative analysis is very well detailed, some minor comments could improve the quality of the manuscript.

1.     In supplementary table 1, the etiology of the liver disease of the 9 patients included and the BCLC classification of each of them should be clear and specified. It is not clear and it is something important to describe.

2.     Among the regulatory mechanisms of cell death, only the vesicular proteins linked to apoptosis are discussed. I would like to know if you have identified any vesicular protein linked to ferroptosis, senescence, necrosis, autophagy?

3.     The first paragraph of the discussion can be completely eliminated, it is repetitive of what was discussed about EVs in the introduction. The discussion should begin with the main findings of the manuscript, paragraph two.

4.     A great limitation is the very small N, however the great strength of the study is the application of the SILAC methodology. It would be important to add to the strengths the time required by this type of methodology compared to traditional ones and finally hypothesize in the discussion about the translation into clinical practice, possibilities or how far we are.

5.     Overall, the spelling and grammar are incorrect, so we recommend that you get the English proofread.

Author Response

Please see the attachment. Note: Response to all 3 reviewers is attached; Data material adressing comment 2 of reviewer 1 is uploaded as zip file as well (contains 2 excel tables)

Reviewer 2 Report

Comments and Suggestions for Authors

Resch et al. described a proteomics study in samples prepared with ultracentrifugation to isolate extracellular vesicles from patient-derived plasma. The study aimed to address the questions related to post-hepatectomy liver failure. In general, the manuscript may poses research interests to the audience of IJMS, however further improvements are needed and major revisions are required on the manuscript

1. While the authors emphasize that the study focuses on plasma EVs, data demonstrating the purity of the isolated EVs is lacking. Furthermore, defining the quantified proteins in this study as ‘EV proteins’ may be misleading. Many of the identified proteins, such as albumin and apolipoproteins, are likely contaminants from plasma rather than EV proteins. The prominent presence of albumin in the EVs suggests significant plasma protein contamination in the EV preparation. A further data clean-up may be performed to identify EV proteins before bioinformatic analysis.

2. Line 384. The authors claimed that 'Reproducibility of ultracentrifugation-based EV isolation had been proven beforehand with 3 healthy subjects and is available upon request'. However, since the reproducibility and composition of EVs obtained by ultracentrifugation are affected by the experimental parameters in ultracentrifugation, such data is an important part of demonstrating the validity of an EV proteomics study and should be included as a part of supplementary data. Otherwise, proper references for the adopted method should be cited.

3. Line 126. The practice of excluding ‘proteins detectable in only one condition’ requires further justification. It is possible that such proteins are condition-specific and may contain information of the biological status. More exploration on the nature of missing data is needed before manipulating the dataset.

4. Line 159. While the authors performed a protein interaction network analysis, no discussion was made in this regard.

5. The authors provided a comprehensive description of the biological functions of the proteins. However, the manuscript does not address the relationship between these proteins and their relationship/indication to PHLF or ALPPS, which is crucial for interpreting the data and answering the questions in this study. 

6. Line 251. The authors suggested several significantly altered proteins as potential predictive signatures of PHLF based on the separation of study groups in the PCA plot. However, it is important to note that such results are insufficient to identify these proteins as predictive signatures, as the PCA reflects the differences between groups rather than predicting the difference or outcome. 

7. The study design description may lack sufficient detail. For instance, the exact meaning and the role of the ‘ALPPS’ group in this study and the effects of this procedure on patients are unclear.

8. The manuscript contains many abbreviations (except for gene names), making it difficult for the audience to follow. For example, the abbreviations used for sampling time points, such as PRE, POD1, and POD5, are unnecessarily complex. Using simpler terms like Day 1 and Day 5 would be more understandable for the journal’s readership.

9. Arguments in the manuscript need to be carefully refined. For example, in the Abstract part, it is inappropriate to suggest 'EV as potential biomarkers' since the study focus on proteins; the authors claimed the results supporting the establishments of potential biomarkers for therapeutic targets, however the results and discussion in this manuscript are insufficient to support this arguments. 

Comments on the Quality of English Language

The quality of English Language is ok.

Reviewer 3 Report

Comments and Suggestions for Authors

The authors aimed to isolate plasma nanoparticles (extracellular vesicles) that could be indicative for liver failure due to formerly performed partial hepatectomy. The study was performed on nine patients with HCC. The topic is novel and important from a clinical point of view. The methodology is well described and supported by detailed figures. Nevertheless I have several suggestions:

  1. In the last part of the introduction, please, specify the aim of the survey.

  2. Point ‘2’ should be: Materials and methods instead of Results.

  3. The original point 4. Materials and Methods should be transferred - for now it is mistakenly placed as the last point.

  4. The last paragraph of the discussion should be changed in order to strengthen the impact of the survey on science and clinics. It should also be   moved to the conclusions.

Round 2

Reviewer 2 Report

Comments and Suggestions for Authors

The authors have addressed my questions. I would suggest accepting the manuscript.